# Risk Factors and Prognosis of Stroke in Gynecologic Cancer Patients

**DOI:** 10.3390/cancers15194895

**Published:** 2023-10-09

**Authors:** Ji Young Kwon, Kena Park, Jeong Min Song, Seung Yeon Pyeon, Seon Hwa Lee, Young Shin Chung, Jong-Min Lee

**Affiliations:** 1Department of Obstetrics and Gynecology, Kyung Hee University Hospital at Gangdong, Seoul 05278, Republic of Korea; kwonji33@gmail.com (J.Y.K.); kenapark2013@gmail.com (K.P.); jeongsmile@gmail.com (J.M.S.); pyun0522@gmail.com (S.Y.P.); bono03@naver.com (Y.S.C.); 2Department of Medicine, Graduate School, Kyung Hee University, Seoul 02447, Republic of Korea; 3Medical Big Data Research Center, Research Institute of Clinical Medicine, Kyung Hee University Hospital at Gangdong, Seoul 05278, Republic of Korea; sproutlife@naver.com

**Keywords:** gynecologic cancers, stroke, risk, prognosis

## Abstract

**Simple Summary:**

The impact of stroke on patients with cancer is of significant concern, as gynecologic oncologists occasionally face medical challenges in managing patients who experience stroke during cancer treatment. However, the relationship between stroke and gynecological cancers has not been sufficiently explored. Here, we evaluated the risk factors for stroke in patients with three major types of gynecological cancers and their effects on cancer prognosis. Out of the 644 patients, 54 (8.4%) experienced a stroke, which is notably higher than the 2% of stroke occurrence rate in the general population. Stroke is significantly associated with age and hypertension. Contrary to our initial concerns, stroke was not found to be an independent prognostic factor for progression-free or overall survival. These findings suggest the importance of an appropriate management plan that considers the age, medical history, and tumor characteristics of the patient, rather than an excessive concern about stroke itself.

**Abstract:**

Increased life expectancy and cancer prevalence rates expose patients to a higher risk of developing other comorbidities such as stroke. This study aimed to evaluate the risk factors for and prognosis of stroke in patients with gynecological cancers. A single-center retrospective cohort study was conducted on patients with cervical, endometrial, and epithelial ovarian cancers. Patients were classified into three groups based on the period of stroke onset: at least one year before cancer diagnosis, within one year before cancer diagnosis to six months after the last treatment date, and six months after the last treatment date. Among the 644 patients, stroke occurred in 54 (8.4%). In univariate analysis, stroke was significantly associated with overall survival. In contrast, in multivariate analysis, stroke was significantly associated with age and hypertension, but not with overall survival. Age, pulmonary thromboembolism/deep vein thrombosis, histological grade, and tumor stage were significantly associated with overall survival. Therefore, it is important to establish an appropriate examination and treatment plan for patients with gynecologic cancers using a multidisciplinary approach that incorporates the patient’s age, medical condition, and tumor characteristics rather than excessively considering the adverse effects of stroke on cancer prognosis.

## 1. Introduction

According to Statistics Korea, Korean society has been aging since 2000, with the population of over 65-year-olds exceeding 7% of the total population. In particular, the life expectancy of Korean women has improved from 65.8 years in 1970 to 86.6 years in 2021, resulting in a shift in the incidence of common medical diseases and causes of death [1]. In 2020, the leading causes of death among Korean women were cancer followed by heart disease, stroke, pneumonia, intentional self-harm, diabetes mellitus (DM), Alzheimer’s disease, liver disease, hypertensive disease, and sepsis. Importantly, the mortality rate due to pneumonia has steadily decreased. Thus, the order of stroke and pneumonia was reversed compared with previous data [2]. As living longer has inherent risks of developing cancer, the predicted probability of developing cancer upon surviving to life expectancy was to be 33.9% in 2021 [1,3]. 

In Korea, more than 100,000 people suffer from new or recurrent stroke every year. With increasing age, the stroke incidence dramatically increases from 0.02% in the population aged ≤44 years to 3.3% in those aged ≥85 years [4]. By 2030, the annual stroke incidence in Korea is estimated to increase to 350,000 resulting from a rapid increase in the elderly population [1]. 

Cancer has been recognized as a cause of stroke because of its hypercoagulability, non-hypercoagulable mechanisms, direct tumor effects, and cancer treatment-related adverse effects [5]. In addition, considering the common risk factors for cancer and stroke, the number of cancer patients with stroke has been increasing, particularly in the elderly population [6]. Stroke in patients with cancer is expected to become a more critical issue as many oncologists sometimes face medical challenges in managing patients who experience stroke during cancer treatment. For example, cancer patients may not receive standard stroke care, and stroke could also disturb cancer patients from receiving the optimal management. Considering the significant risks of morbidity and mortality associated with both conditions, understanding the relationship between cancer and stroke is important for developing a proper management plan.

In 2020, the number of Korean women newly diagnosed with gynecological cancer exceeded 9400. Over the past two decades, distinct trends in cancer incidence by cancer type have been observed [7,8,9,10]. As screening for cervical cancer has become widely accepted and interest in HPV vaccines has increased, the incidence of cervical cancer has declined in recent decades [11,12]. However, the incidence of endometrial and ovarian cancers has dramatically increased due to the increased prevalence of obesity and prolonged life expectancies [13,14]. Advancements in medicine and the easily accessible healthcare system in Korea have led to an increase in the number of newly diagnosed gynecologic cancer patients, but also the prevalence of gynecologic cancers. This increase in life expectancy and cancer prevalence rates leads to a concomitant increase in the risk of developing other comorbidities such as stroke [15,16,17]. 

Previous studies have suggested a relationship between stroke and other solid cancers such as lung, prostate, breast, colorectum, brain, and lymphoma [18,19]. However, the association between stroke and gynecologic cancers remains underexplored [20]. Thus, this study aimed to investigate the association between the three major types of gynecological cancer and stroke and evaluate the risk factors and prognosis of stroke in patients with cancer.

## 2. Materials and Methods

This single-center retrospective cohort study was conducted on patients with cervical, endometrial, and epithelial ovarian cancers from May 2006 to May 2022. This study was approved by the Institutional Review Board of Kyung Hee University Hospital at Gangdong, Seoul, Korea (IRB no: 2022-07-043). Exclusion criteria included patients who did not receive definitive cancer treatment, those who were transferred from or out to other clinics during treatment, and those who were diagnosed with cancer types other than cervical, endometrial, and epithelial ovarian cancers.

Clinicopathological data such as age and body mass index (BMI) at the time of cancer diagnosis, medical history, smoking history, cancer type, histological subtype and grade, International Federation of Gynecology and Obstetrics (FIGO) stage, and type of cancer treatment were collected. Considering the prognostic impact of histological subtypes according to cancer origin, the usual histology was defined as squamous carcinoma or adenocarcinoma in the cervix, endometrioid carcinoma in the endometrium, and serous or endometrioid carcinoma in the ovary. An unusual histology was defined as a pathologically confirmed cell type other than the usual histological type of each cancer.

Based on preliminary data reviewed in previous studies and considering the natural history of stroke, patients with stroke were classified into three groups [15,21,22]. Patients who had experienced an attack at least one year before the date of cancer diagnosis were grouped into the stroke group before treatment. Patients diagnosed with stroke within one year before the date of cancer diagnosis to six months from the last date of treatment were grouped into the stroke group during treatment. Patients who experienced the attack six months after the last date of treatment were grouped into the stroke group after treatment.

The baseline characteristics of the groups were compared using descriptive statistics. The *t*-test or Mann–Whitney *U* test and chi-square or Fisher’s exact test were used for the continuous and categorical variables, respectively. Risk factors of stroke in gynecologic cancers were evaluated using a logistic regression model, and variables with *p*-values < 0.05 in univariate analysis were further analyzed using multivariate analysis. Overall survival (OS) and progression-free survival (PFS) were compared between groups using Kaplan–Meier survival curves and log-rank tests. The prognostic factors for OS and PFS were evaluated using the Cox proportional hazards model and presented as hazard ratios (HR) and 95% confidence intervals (CI). Data were analyzed using IBM SPSS version 25 for Windows (IBM Corp., Armonk, NY, USA). Statistical significance was set at *p* < 0.05.

## 3. Results

A total of 1069 patients with gynecological cancer were enrolled in this study. Of these, 98 patients were excluded because they did not receive definitive cancer treatment; 290 were excluded because they were transferred from or out to other hospitals; 37 were excluded because of cancer types other than cervical, endometrial, and epithelial ovarian cancers. Of the 644 patients included, 54 (8.4%) were assigned to the stroke group and 590 (91.6%) to the non-stroke group. Patients who experienced stroke were divided into three subgroups according to stroke onset: before cancer treatment (29 patients), during treatment (19 patients), and after treatment (six patients) (Figure 1).

### 3.1. Baseline Characteristics

The baseline characteristics of individuals with or without stroke are shown in Table 1. The median age of the participants was 54 years (range: 20–89 years). The age at cancer diagnosis was higher in the stroke group than in the non-stroke group (*p* < 0.001). The median BMI, pulmonary thromboembolism/deep vein thrombosis (PTE/DVT), fractures, and smoking status were not significantly different between the stroke and non-stroke groups. Common comorbidities such as DM, hypertension (HTN), and heart disease were more frequently observed in the stroke group than in the non-stroke group (*p* < 0.001). The number of patients diagnosed with cervical, endometrial, and epithelial ovarian cancers were 257, 179, and 208, respectively; however, no significant difference was observed between the stroke and non-stroke groups. In addition, important prognostic factors such as histological subtype, grade, and FIGO stage showed no significant differences between the groups. Surgery was performed more frequently in the non-stroke group than in the stroke group (*p* = 0.01). However, the percentages of patients who received radiation therapy or chemotherapy were comparable between the groups.

### 3.2. Risk Factors of Stroke in Patients with Cancers

Various clinicopathological variables were evaluated using logistic regression models to identify the risk factors for stroke (Table 2). In univariate analysis, age (odds ratio [OR], 1.074; 95% CI, 1.050–1.098; *p* < 0.001), DM (OR, 3.597; 95% CI, 1.941–6.668; *p* < 0.001), HTN (OR, 7.426; 95% CI, 3.982–13.850; *p* < 0.001), and heart disease (OR, 4.141; 95% CI, 2.296–7.470; *p* < 0.001) were significantly associated with the increased risk of stroke. Surgical treatment (OR, 0.411; 95% CI, 0.205–0.822; *p* = 0.012), however, was significantly less likely to be associated with stroke risk. Contrary to expectations, BMI, PTE/DVT, fracture, smoking history, cancer type, histologic subtype, tumor grade, FIGO stage, radiotherapy, and chemotherapy were not associated with the risk of stroke, even in univariate analysis. In multivariate analysis, patient age at cancer diagnosis (OR, 1.055; 95% CI, 1.027–1.084; *p* < 0.001) and HTN (OR, 3.579; 95% CI, 1.801–7.115; *p* < 0.001) were identified as significant risk factors for stroke.

### 3.3. Prognosis of Patients with Cancer and Stroke

The prognosis of patients with gynecologic cancer and stroke was evaluated using three-year PFS and overall survival (OS) using the Kaplan–Meier curve with the log-rank test. Stroke did not significantly affect PFS, even in patients during treatment (HR, 1.714; 95% CI, 0.961–3.056; *p* = 0.068) as well as before or after treatment (HR, 1.289; 95% CI, 0.809–2.051; *p* = 0.285) (Appendix A, Appendix A). However, OS was significantly lower in the stroke group than in the non-stroke group (HR, 2.789; 95% CI, 1.859–4.185; *p* < 0.001), especially in patients with stroke during treatment (HR, 4.279; 95% CI, 2.501–7.321; *p* < 0.001) (Figure 2). Using the Cox proportional hazards model, stroke was evaluated as an independent prognostic factor of OS in the study population. In multivariate analysis, stroke during treatment (HR, 2.182; 95% CI, 0.903–5.272; *p* = 0.083) as well as before and after stroke (HR, 2.030; 95% CI, 0.856–4.816; *p* = 0.108) was not significantly associated with OS. Instead, age, PTE/DVT, histological grade, and FIGO stage were significantly associated with OS (Table 3).

## 4. Discussion

Prolonged life expectancies and improved survival rates due to advancements in cancer treatment have raised concerns about patient care, as these can increase the risk of medical comorbidities such as stroke. The coexistence of cancer and stroke could not only be an emotional threat to patients and their families, but also a real burden on physicians. Moreover, this may be a hurdle that precludes patients from receiving standard care for both cancer and stroke. Thus, it is important to understand the interaction between cancer and stroke on the risk of occurrence and prognosis as well as the clinical characteristics of each patient. This study examined the risk factors for stroke in patients with gynecological cancer and the effects of stroke on cancer prognosis.

The risk of stroke is increased in patients with cancer. An alarming prevalence of stroke in patients with cancer has been reported to be as high as 15%, although it may be approximately 20% in some Asian populations [23]. Among these, hematological malignancies in childhood and adenocarcinoma of the lungs, colorectum, and pancreas in adults are the most commonly identified. Five to ten percent of patients with acute stroke present with an active cancer, whether or not previously identified, and another 3 to 5% receive a new cancer diagnosis during the two years after their ischemic stroke [6,21,24,25,26]. In this study, among the 644 patients with gynecological cancer, 54 (8.4%) experienced stroke, which was higher than that in the general population previously reported in Korea. According to Stroke Statistics in Korea 2018, the overall prevalence of stroke in the general female population was 2.56% [27]. This high incidence may reflect the demographic and disease-specific characteristics of the disease. The well-known risk factors for stroke include old age, HTN, obesity, DM, cardiovascular diseases, fractures, and smoking, which coincide with cancer risk factors [21,28,29]. As with our findings, old age and HTN have been reported to be the most significant risk factors for stroke in cancer and non-cancer patients [21,28,29]. Old age is the most important nonmodifiable risk factor for stroke, doubling every 10 years after the age of 55. Approximately three-quarters of all strokes occur in patients aged more than 65 years [30]. HTN is the most prevalent modifiable risk factor for stroke, accounting for approximately 64% of patients and controlling HTN to <150/90 mmHg reduces the risk of stroke [31,32,33,34]. However, certain risk factors such as BMI, smoking, and fractures were not significantly associated with stroke in the current study. In general, a high BMI has been reported to increase the risk of stroke due to the difficulty in blood flow and an increased risk of blockage with inflammation caused by excess fatty tissue [35,36]. A strong association between smoking and stroke has been reported; current smokers have at least a 2–4 times increased risk of stroke due to atherosclerosis and arterial damage compared to lifelong nonsmokers or individuals who had quit smoking more than 10 years prior [37]. Previous studies have suggested an up to 4-fold increase in the risk of fractures in patients with stroke compared with healthy controls [38,39]. Although the incidence of stroke after bone fracture is rare, history of prior fracture is associated with a 6.4-fold increase in the risk of post fracture ischemic stroke [40,41]. The negative findings of this study are probably because, compared to Western countries, the mean BMI of Korean women is much lower, especially in older patients, and few of them are smokers [42,43]. Since many elderly women whose movement is restricted by fractures choose palliative management over definitive treatments, these patients were eliminated from the study group, resulting in an insignificant stroke incidence.

Moreover, cancer itself seems to be implicated in stroke pathophysiology as it increases the risk of stroke [6,16,17,23,28]. Cancer-related complications that may cause stroke include brain metastases and cerebral emboli. A cancerous mass may cause stroke by occluding blood flow or inducing thrombosis and vessel spasms [25]. Coagulopathy and infections are other cancer complications that cause thrombosis or hemorrhage by damaging the endothelium of cerebral blood vessels [44]. Cancer may also be associated with stroke due to therapeutic interventions. Radiotherapy can damage and cause atherosclerotic changes in cerebral blood vessels [26]. Platinum compounds, which are most commonly used in gynecological cancers, seem to present a significant risk for stroke. Although the mechanisms involved remain largely unknown, platinum-associated thrombosis is theorized to be the result of direct endothelial damage and reduced anticoagulant factor synthesis [25,44,45]. Although cancer progression coupled with the acute emergence of stroke raises the suspicion of tumor emboli, it is almost impossible to differentiate whether stroke is a result of atherosclerotic or tumor-associated changes [6,25,44]. Similarly, we also did not find an increased possibility of tumor emboli as a direct cause of stroke because histological subtype, grade, and FIGO stage did not contribute to an increase in stroke risk.

The effect of stroke on cancer prognosis was also evaluated. Stroke can increase the burden of treatment and negatively affect the prognosis of cancer patients. In a large cohort study using the National Cancer Institute’s Surveillance, Epidemiology, and End Results program, patients with cancer were more than twice as likely to die of a stroke than the general population, and the risk increased with time. Additionally, cancers of the breast, prostate, or colorectum are most associated with fatal stroke [45,46]. As expected, in univariate analysis, stroke during treatment had an adverse effect on PFS. Stroke during treatment as well as before and after treatment also showed significant adverse effects on OS. However, in multivariate analysis, stroke did not convey clinical significance to the PFS and OS of patients with gynecologic cancer. Instead, age, PTE/DVT, histological grade, and FIGO stage were identified as independent prognostic factors in this study population. These findings might reflect the different demographic characteristics, which were previously mentioned, compared to Western populations and the favorable tumor biology of gynecologic malignancies. Cervical and endometrial cancers, which account for two-thirds of the study population, are characterized by a relatively early stage diagnosis and good prognosis compared with other non-gynecologic malignancies [9,11,13,43]. Additionally, the fact that stroke was not identified as an independent prognostic factor in our cancer patients may imply an advanced and well-organized healthcare system in Korea [47]. As advancements in modern medicine have improved the survival and life expectancy of patients with cancer, the incidence rate of stroke has also increased [48]. Moreover, developments in imaging studies such as magnetic resonance imaging (MRI) with improved health insurance coverage have led to the early diagnosis of stroke and medical interventions, resulting in improved mortality rates of 12.8% from 2014 to 2019 [47,49,50]. Therefore, the prognoses of gynecologic cancer patients with stroke are not significantly associated with the occurrence of stroke, but rather with the patients’ age and histopathological characteristics of the cancers.

In this study, patients were managed using a multidisciplinary team approach. Patient medical histories such as obesity, HTN, PTE/DVT, and history of stroke or fractures as well as physical examinations were thoroughly reviewed, followed by appropriate imaging and treatment modalities including brain MRI, computed tomography (CT), DVT ultrasonography, and antithrombotic treatment, if needed. Patients who had experienced a stroke at least one year prior to cancer diagnosis were treated with standard cancer treatment. Additionally, patients who experience neurological symptoms during cancer treatment undergo an immediate diagnostic process for stroke and receive appropriate treatment. Consequently, no patient loss could be directly attributed to stroke during cancer treatment.

## 5. Conclusions

Although this study was limited by its single-institute retrospective design and relatively small sample size, to our knowledge, this is the first report on the relationship between cancer and stroke in patients with the three major types of gynecologic cancers. Old age and HTN were found to be significant risk factors for stroke in patients with gynecological cancer. Stroke did not adversely affect the prognosis of gynecological cancers. Thus, it is crucial to establish an appropriate examination and management plan by adopting a multidisciplinary approach that incorporates the patient age, medical disease, and tumor characteristics rather than excessively considering the adverse effects of stroke on cancer prognosis.

## Figures and Tables

**Figure 1 cancers-15-04895-f001:**
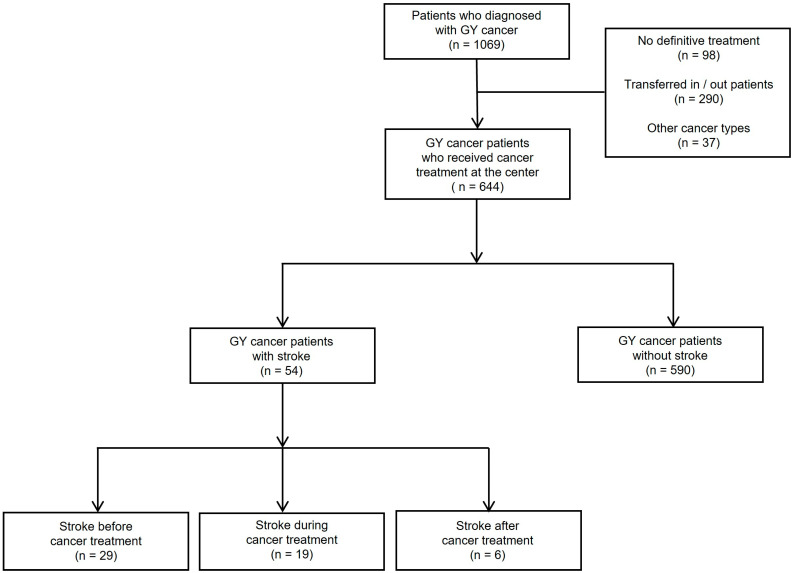
Enrollment and classification of stroke groups by the diagnostic onset of stroke relative to the period of cancer treatment.

**Figure 2 cancers-15-04895-f002:**
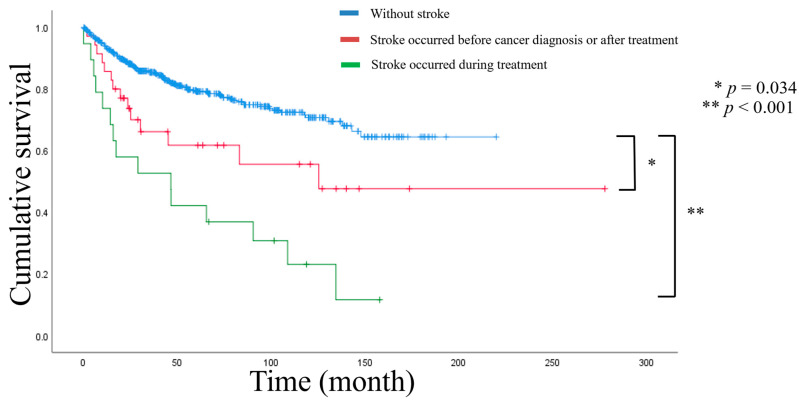
Kaplan–Meier curves of overall survival by stroke onset in comparison to the non-stroke group.

**Table 1 cancers-15-04895-t001:** Patients’ characteristics at the baseline.

Characteristics	With Stroke (*n* = 54)	Without Stroke (*n* = 590)	*p*-Value
Median age at diagnosis, years (range)	65.50 (36–86)	53 (20–89)	<0.001
Median BMI at diagnosis, kg/m^2^ (range)	25 (16–36)	24 (13–40)	0.219
Comorbidities, *n* (%)			
DM	18 (33.3)	72 (12.2)	<0.001
HTN	39 (72.2)	153 (25.9)	<0.001
PTE/DVT	5 (9.3)	25 (4.2)	0.097
Heart disease ^1^	22 (40.7)	84 (14.2)	<0.001
Fractures ^2^	4 (7.4)	19 (3.2)	0.118
Smoking	3 (5.6)	36 (6.1)	>0.999
Cancer type			0.805
Cervix	21 (38.9)	236 (40.0)	
Endometrium	17 (31.5)	162 (27.5)	
Ovary	16 (29.6)	192 (32.5)	
Histological subtype, *n* (%)			0.289
Usual type ^3^	39 (72.2)	463 (78.5)	
Unusual type	15 (27.8)	127 (21.5)	
Histological grade, *n* (%)			0.170
Well/Moderate	18 (58.1)	274 (69.9)	
Poor	13 (41.9)	118 (30.1)	
FIGO stage, *n* (%)			0.108
I/II	33 (61.1)	422 (71.5)	
III/IV	21 (38.9)	168 (28.5)	
Type of treatment			
Surgery, *n* (%)	42 (77.8)	528 (89.5)	0.010
Radiotherapy, *n* (%)	15 (27.8)	130 (22.0)	0.333
Chemotherapy, *n* (%)	31 (57.4)	304 (51.5)	0.408

^1^ Heart disease includes atrial fibrillation, coronary artery disease, dyslipidemia, or heart failure. ^2^ Fractures include fractures located in the femur, fibular, patella, rib, or spine and total hip or knee replacement. ^3^ Usual type indicates squamous cell or adenocarcinoma in the cervix, endometrioid carcinoma in the endometrium, and serous or endometrioid carcinoma in the ovary. Abbreviations: BMI, body mass index; DM, diabetes mellitus; HTN, hypertension; PTE, pulmonary thromboembolism; DVT, deep vein thrombosis; FIGO, International Federation of Gynecology and Obstetrics.

**Table 2 cancers-15-04895-t002:** Risk factors of stroke in gynecologic cancer patients.

Variables	Univariate Analysis		Multivariate Analysis	
	OR (95% CI)	*p*-Value	OR (95% CI)	*p*-Value
Age (yr)	1.074 (1.050–1.098)	<0.001	1.055 (1.027–1.084)	<0.001
BMI (kg/m^2^)	1.042 (0.977–1.112)	0.210		
DM		<0.001		0.077
No (*n* = 554)	1.00 (Reference)		1.00 (Reference)	
Yes (*n* = 90)	3.597 (1.941–6.668)		1.852 (0.934–3.672)	
HTN		<0.001		<0.001
No (*n* = 452)	1.00 (Reference)		1.00 (Reference)	
Yes (*n* = 192)	7.426 (3.982–13.850)		3.579 (1.801–7.115)	
PTE/DVT		0.103		
No (*n* = 614)	1.00 (Reference)			
Yes (*n* = 30)	2.306 (0.845–6.291)			
Heart disease ^1^		<0.001		0.144
No (*n* = 538)	1.00 (Reference)		1.00 (Reference)	
Yes (*n* = 106)	4.141 (2.296–7.470)		1.635 (0.845–3.164)	
Fractures ^2^		0.124		
No (*n* = 621)	1.00 (Reference)			
Yes (*n* = 23)	2.404 (0.787–7.341)			
Smoking history		0.872		
No (*n* = 605)	1.00 (Reference)			
Past/Current (*n* = 39)	0.905 (0.269–3.042)			
Cancer type		0.661		
Cervix/Endometrium (*n* = 436)	1.00 (Reference)			
Ovary (*n* = 208)	0.873 (0.475–1.605)			
Histological subtype		0.291		
Usual type ^3^ (*n* = 502)	1.00 (Reference)			
Unusual type (*n* = 142)	1.402 (0.749–2.625)			
Histological grade		0.174		
Well/Moderate (*n* = 292)	1.00 (Reference)			
Poor (*n* = 131)	1.677 (0.794–3.519)			
Stage		0.142		
I/II (*n* = 455)	1.00 (Reference)			
III/IV (*n* = 189)	1.529 (0.796–3.534)			
Surgery		0.012		0.984
No (*n* = 74)	1.00 (Reference)		1.00 (Reference)	
Yes (*n* = 570)	0.411 (0.205–0.822)		0.991 (0.420–2.338)	
Radiotherapy		0.335		
No (*n* = 499)	1.00 (Reference)			
Yes (*n* = 145)	1.361 (0.727–2.546)			
Chemotherapy		0.408		
No (*n* = 309)	1.00 (Reference)			
Yes (*n* = 335)	1.268 (0.722–2.227)			

^1^ Heart disease includes atrial fibrillation, coronary artery disease, dyslipidemia, or heart failure. ^2^ Fractures include fractures located in the femur, fibular, patella, rib, or spine and total hip or knee replacement state. ^3^ Usual type indicates squamous cell or adenocarcinoma in the cervix, endometrioid carcinoma in the endometrium, and serous or endometrioid carcinoma in the ovary. Abbreviations: BMI, body mass index; DM, diabetes mellitus; HTN, hypertension; PTE, pulmonary thromboembolism; DVT, deep vein thrombosis.

**Table 3 cancers-15-04895-t003:** Prognostic factors associated with overall survival.

Variables	Univariate Analysis		Multivariate Analysis	
	HR (95% CI)	*p*-Value	HR (95% CI)	*p*-Value
Age (yr)	1.044 (1.031–1.057)	<0.001	1.025 (1.006–1.045)	0.008
BMI (Kg/m^2^)	0.919 (0.879–0.960)	< 0.001	0.952 (0.893–1.015)	0.135
DM				
No	1.00 (Reference)			
Yes	1.287 (0.843–1.965)	0.242		
HTN				
No	1.00 (Reference)		1.00 (Reference)	
Yes	1.433 (1.031–1.993)	0.032	0.706 (0.406–1.227)	0.217
PTE/DVT				
No	1.00 (Reference)		1.00 (Reference)	
Yes	3.166 (1.853–5.410)	<0.001	2.932 (1.430–6.012)	0.003
Heart disease ^1^				
No	1.00 (Reference)			
Yes	1.196 (0.797–1.793)	0.388		
Fracture ^2^				
No	1.00 (Reference)			
Yes	1.504 (0.766–2.954)	0.236		
Stroke onset				
No stroke	1.00 (Reference)		1.00 (Reference)	
Stroke before/after tx	2.027 (1.165–3.528)	0.012	2.030 (0.856–4.816)	0.108
Stroke during tx	4.279 (2.501–7.321)	<0.001	2.182 (0.903–5.272)	0.083
Cancer type				
Cervix/Endometrium	1.00 (Reference)		1.00 (Reference)	
Ovary	3.145 (2.277–4.343)	<0.001	1.606 (0.958–2.693)	0.072
Histological subtype				
Usual ^3^	1.00 (Reference)		1.00 (Reference)	
Unusual	2.010 (1.425–2.834)	<0.001	1.539 (0.921–2.571)	0.100
Histological grade				
Well/Moderate	1.00 (Reference)		1.00 (Reference)	
Poor	4.372(2.851–6.705)	<0.001	1.898 (1.142–3.155)	0.013
Stage				
I/II	1.00 (Reference)		1.00 (Reference)	
III/IV	8.333 (5.873–11.823)	<0.001	3.482 (2.006–6.043)	<0.001

^1^ Heart disease includes atrial fibrillation, coronary artery disease, dyslipidemia, or heart failure. ^2^ Fractures include fractures located in the femur, fibular, patella, rib, or spine and total hip or knee replacement. ^3^ Usual type indicates squamous cell or adenocarcinoma in the cervix, endometrioid carcinoma in the endometrium, and serous or endometrioid carcinoma in the ovary. Abbreviations: OS, overall survival; HR, hazard ratio; CI, confidence interval; BMI, body mass index; DM, diabetes mellitus; HTN, hypertension; PTE, pulmonary thromboembolism; DVT, deep vein thrombosis; Tx, treatment.

## Data Availability

No new data were created.

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
