# Peer review of "Risk Factors and Prognosis of Stroke in Gynecologic Cancer Patients"

_cancers, 2023, doi:10.3390/cancers15194895_

Round 1

Reviewer 1 Report

In this paper, the authors evaluated the risk factors for stroke in patients with cervical, endometrial, and epithelial ovarian cancer. The occurrence of stroke in these cancer patients was found to be higher than that in the general population. Stroke is found to be significantly associated with age and hypertension, but not prognostic of overall survival. The addressed question is relevant and interesting to the bioinformatics community. The reviewer has the following comments:

1. If the authors did not test the proportional hazards assumption for Cox regression analysis, they should do so, for instance, using the scaled Schoenfield residuals. Additionally, the model should include covariates like age, sex, and other confounding variables.

2. The p-values need to be corrected for multiple testing (e.g., Tables 1, 2, and 3).

3. The patients with stroke during or before/after cancer treatment have very limited sample sizes (19 and 35, respectively) compared with the patients without stroke (590 in total). This may have greatly harmed the statistical power, and the negative results may not be true negatives. The authors need to address this limitation in the Discussion section.

4. (Minor) Figure 2 seems to be stretched, and says “p=< 0.001” (should be “p < 0.001” or “p ≤ 0.001”).

Author Response

  1. If the authors did not test the proportional hazards assumption for Cox regression analysis, they should do so, for instance, using the scaled Schoenfield residuals. Additionally, the model should include covariates like age, sex, and other confounding variables.

â–¶ We used Cox proportional hazards models, that included covariates you mentioned above, to evaluate the prognostic factors of OS and PFS. The result was summarized in Table 3 and supplementary table 1.

  1. The p-values need to be corrected for multiple testing (e.g., Tables 1, 2, and 3).

â–¶ The p-values from multivariate analysis were summarized in the tables.

  1. The patients with stroke during or before/after cancer treatment have very limited sample sizes (19 and 35, respectively) compared with the patients without stroke (590 in total). This may have greatly harmed the statistical power, and the negative results may not be true negatives. The authors need to address this limitation in the Discussion section.

â–¶ We mentioned it as one of limitations of this study in line 318-9 (highlighted).

  1. (Minor) Figure 2 seems to be stretched, and says “p=< 0.001” (should be “p < 0.001” or “p ≤ 0.001”).

â–¶ We corrected it.

Reviewer 2 Report

The article presented for review is an interesting work, well written and well planned.

Minor notes: The authors in the text (line 164-177, 191-202) repeat the OP values, which are given in the tables below. The text with so many numbers and intervals is difficult to read, I think these paragraphs should be redone.

Author Response

The authors in the text (line 164-177, 191-202) repeat the OP values, which are given in the tables below. The text with so many numbers and intervals is difficult to read, I think these paragraphs should be redone.

â–¶ I deleted some redundant values (highlighted).

Reviewer 3 Report

This research analysis stroke in ginecological cancer patients. The research analyzed a large series of cases, but the number of strokes was "relatively low" (n = 54). This is a limitation of the study that is acknowledged. I am not sure if the authors need to do any king of power analysis or statistical test to confirm if 54 cases is enough. Nevertheless, the results may be of interest to other doctors.

Specific comments:

(1) Regarding Lines 54-55. I do not fully understand the meaning. Could you please rewrite it, or explain it again?

(2) In section 58-68. It is stated that stroke and cancer are related, mainly because of neoplasia-associated hypercoagulability. What about as secondary effect of cancer treatment?

(3) Line 70. Instead of using absolute number such as 9,400. Is it possible to transform into percentages? Depending on the total number of women of Korea the percentage will change, and it may be more informative than the absolute value.

(4) Line 79, can you please also describe the principal causes of stroke?

(5) Lines 103-109. Are the time definitions of the 3 groups an standard definition, also used in other publications?

(6) Line 116. Could you please define OS and PFS? From what time, to what event?

(7) Line 125, the final series is comprised of 54 and 590 cases. With 54 cases, do you have enough statistical power?

(8) Lines 164-177. Since the same data is in Table 2. Why not just leave the most relevant?

(9) When comparing with and without stroke, have you considered doing a Propensity score matching (PSM)-style analysis.

(10) What factors are associated with the "stroke occurred during treatment" group that is the one with worse overall survival? Is the bad survival due to the stroke, cancer evolution, or comorbidity?

Author Response

(1) Regarding Lines 54-55. I do not fully understand the meaning. Could you please rewrite it, or explain it again?

â–¶ The incidence of stroke in populations by ages was described in crude incidence rate. However, we changed the expression to improve readers’ understanding as follows: With increasing age, the stroke incidence dramatically increased from 0.02% in population aged ≤ 44 years to 3.3% in those aged ≥ 85 years (highlighted)

(2) In section 58-68. It is stated that stroke and cancer are related, mainly because of neoplasia-associated hypercoagulability. What about as secondary effect of cancer treatment?

â–¶ The effects of cancer or cancer treatment as a cause of stroke were described in line 261-74 (highlighted).

(3) Line 70. Instead of using absolute number such as 9,400. Is it possible to transform into percentages? Depending on the total number of women of Korea the percentage will change, and it may be more informative than the absolute value.

â–¶ About 9,400 persons were newly diagnosed as gynecologic cancers in 2020. However, this is only 0.04% of all Korean women. This can lead to misunderstanding as the incidence of gynecological cancer is low. So, we maintained the original description.

(4) Line 79, can you please also describe the principal causes of stroke?

We described the risk factors (old age, HTN, obesity, DM, cardiovascular diseases, fractures, and smoking) of stroke in line 237-9, and discussed their significance on our study population in section of discussion (highlighted).  

(5) Lines 103-109. Are the time definitions of the 3 groups an standard definition, also used in other publications?

â–¶ Yes. The definitions were used in other publications such as Cancer-Related Stroke: an Emerging Subtype of Ischemic Stroke with Unique Pathomechanisms and Prior Cancer in Patients with Ischemic Stroke: the Bergen NORSTROKE Study.

(6) Line 116. Could you please define OS and PFS? From what time, to what event?

â–¶ OS was defined as the time from histologic diagnosis to death from any cause.

PFS was defined as the time from histologic confirmation to the earliest occurrence of progression on histologic and/or imaging studies.

â–¶ However, these definitions are not described in the text because they are consistent with commonly used concepts.

(7) Line 125, the final series is comprised of 54 and 590 cases. With 54 cases, do you have enough statistical power?

â–¶ We agree. So, we mentioned it as one of limitations of this study in line 318-9 (highlighted).

(8) Lines 164-177. Since the same data is in Table 2. Why not just leave the most relevant?

â–¶ We revised according to your comments (highlighted).

(9) When comparing with and without stroke, have you considered doing a Propensity score matching (PSM)-style analysis.

â–¶ Yes. We have considered PSM analysis, but we did not use it. It reduced the total sample size and no statistical finding is earned by matching a variable.

(10) What factors are associated with the "stroke occurred during treatment" group that is the one with worse overall survival? Is the bad survival due to the stroke, cancer evolution, or comorbidity?

â–¶ In multivariate analysis, stroke during treatment as well as before and after stroke was not significantly associated with OS. Instead, age, PTE/DVT, histological grade, and FIGO stage were significantly associated with OS: Line 196-202 (highlighted).